# Pristine oceans are a significant source of uncertainty in quantifying global cloud condensation nuclei

Goutam Choudhury[1], Karoline Block[2], Mahnoosh Haghighatnasab[2,3], Johannes Quaas[2], Tom Goren[1], and Matthias Tesche[2]

[1]Department of Environment, Planning and Sustainability, Bar-Ilan University, Ramat Gan, 5290002, Israel
[2]Leipzig Institute for Meteorology, Leipzig University, Leipzig, 04103, Germany
[3]Research and Development, German Weather Service, Offenbach am main, 63067, Germany

**Correspondence:** Goutam Choudhury (goutam.choudhury@biu.ac.il)

**Abstract.** Quantifying global cloud condensation nuclei (CCN) concentrations is crucial for reducing uncertainties in radiative forcing resulting from aerosol-cloud interactions. This study analyzes two novel, independent, open-source global CCN datasets derived from spaceborne Cloud Aerosol Lidar with Orthogonal Polarization (CALIOP) measurements and Copernicus Atmosphere Monitoring Service (CAMS) reanalysis and examines the spatio-temporal variability of CCN concentrations pertinent to liquid clouds. The results reveal consistent large-scale patterns in both CALIOP and CAMS datasets, although CALIOP values are approximately 79 % higher than those from CAMS. Comparisons with existing literature demonstrate that these datasets effectively bound regionally observed CCN concentrations, with CALIOP typically representing the upper bound and CAMS the lower bound. Monthly and annual variations in CCN concentrations obtained from the two datasets largely agree over the Northern Hemisphere and align with previously reported variations. However, inconsistencies emerge over pristine oceans, particularly in the Southern Hemisphere, where the datasets show not only opposing seasonal changes but also contrasting annual trends. Seasonal cycles in these regions are well represented in CAMS, consistent with previous in-situ observations, while annual trends seems to be better captured by CALIOP. A closure study of trends in CCN and cloud droplet concentrations suggests that dust-influenced and pristine-maritime environments primarily limit our current understanding of CCN-cloud-droplet relationships. Long-term CCN observations in these regions are crucial for improving global datasets and advancing our understanding of aerosol-cloud interactions.

## 1 Introduction

Aerosols act as cloud condensation nuclei (CCN) and through aerosol–cloud interactions (ACIs) induce a cooling effect on the climate, partially offsetting the warming due to greenhouse gases (Forster et al., 2021). The effective radiative forcing due to ACIs (ERF$_{ACI}$) is however highly uncertain, estimated to range between -1.7 and -0.3 W m$^{-2}$ with moderate confidence (Forster et al., 2021).

A fundamental parameter for constraining ERF$_{ACI}$ is the number concentration of CCN forming aerosols ($n_{CCN}$). Satellite-based studies of ERF$_{ACI}$ rely on aerosol optical properties as proxies for $n_{CCN}$. Part of the uncertainty in ERF$_{ACI}$ arises from variations in estimates between different observation-based reports, particularly due to their choice of $n_{CCN}$ proxy (Forster

et al., 2021; Gryspeerdt et al., 2017). The most common proxies are aerosol optical depth (AOD) and aerosol index (AI)
(Quaas et al., 2020; Rosenfeld et al., 2023). AOD, being a column-integrated bulk property, poorly represents $n_{CCN}$ at cloud
level. AI, calculated from AOD and Ångström exponent, gives more weight to fine particles and offers an improvement over
AOD. Using AI over AOD strengthens the negative radiative forcing by at least 30 % (Gryspeerdt et al., 2017). Nevertheless,
because AI is derived from AOD, it inherits the limitations of AOD (Quaas et al., 2020; Rosenfeld et al., 2023). Incorporating
additional polarimetric measurements enables retrievals of atmospheric-column-integrated aerosol number concentrations over
oceans, which have been shown to yield a significantly stronger negative forcing compared to AOD and AI (Hasekamp et al.,
2019). Despite being a significant improvement over optical proxies, these concentrations are still column-integrated and may
not represent the cloud-level values most relevant to ACIs. These studies illustrate that $ERF_{ACI}$ significantly varies with the
choice of $n_{CCN}$ proxy and highlight the critical need for a comprehensive, height-resolved global $n_{CCN}$ dataset as the next
essential step for advancing $ERF_{ACI}$ estimates.

Two recent efforts have addressed these limitations. Choudhury and Tesche (2023a) present a satellite-derived, vertically-
resolved, three-dimensional (3D) dataset of global $n_{CCN}$. Their approach leverages the Cloud Aerosol Lidar with Orthogonal
Polarization (CALIOP) retrievals and employs a validated CCN-retrieval algorithm (Choudhury and Tesche, 2022a) to retrieve
$n_{CCN}$ from profiles of aerosol extinction coefficient. The retrieved $n_{CCN}$ are then gridded onto a 2º by 5º latitude-longitude
grid with a vertical resolution of 60 m to produce a monthly global $n_{CCN}$ dataset. The robustness of the retrieval algorithm is
established through comparisons with in-situ measurements from various land and ocean-based platforms (Choudhury et al.,
2022; Choudhury and Tesche, 2022b; Aravindhavel et al., 2023).

Complementing this effort, Block et al. (2024) present a 3D global $n_{CCN}$ dataset estimated from the Copernicus Atmosphere
Monitoring Service (CAMS) aerosol reanalysis (Inness et al., 2019a). This dataset is based on a diagnostic box model built on
a simplified Kappa-Köhler framework that estimates $n_{CCN}$ from CAMS-derived aerosol mass mixing ratios. It offers a high
temporal resolution of one day, a horizontal resolution of 0.75º, and a hybrid sigma-pressure vertical grid with 60 levels. While
the validation of this dataset is ongoing, a preliminary comparison to surface-based in-situ observations gives promising results
(Block et al., 2024).

The CAMS $n_{CCN}$ dataset with its high spatio-temporal resolution has great potential for better constraining $ERF_{ACI}$. How-
ever, its dependency on satellite-derived AOD (assimilated into CAMS) and the reliance on modelled aerosol inventories in its
simulated component (Inness et al., 2019a) necessitates an extensive evaluation to assess the representativeness of this dataset.
The CALIOP data's coarse monthly resolution complicates a direct integration into $ERF_{ACI}$ estimation. Nevertheless, it was
found to be representative of in-situ measured long-term variations in $n_{CCN}$ at multiple regional continental sites (Choudhury
and Tesche, 2022b). Thus, the CALIOP $n_{CCN}$ dataset, currently the only satellite-based 3D global data available, presents a
valuable tool for expanding the assessment of the CAMS dataset to a global scale, particularly in regions with limited in-situ
observations.

Here, we conduct a closure study between the two independent novel $n_{CCN}$ datasets, reconciling not only their variability
across diverse spatio-temporal scales but also their co-variability with relevant cloud properties. Furthermore, we augment their
validation by comparing their regional concentrations with in-situ measurements from the literature. The comparative analysis

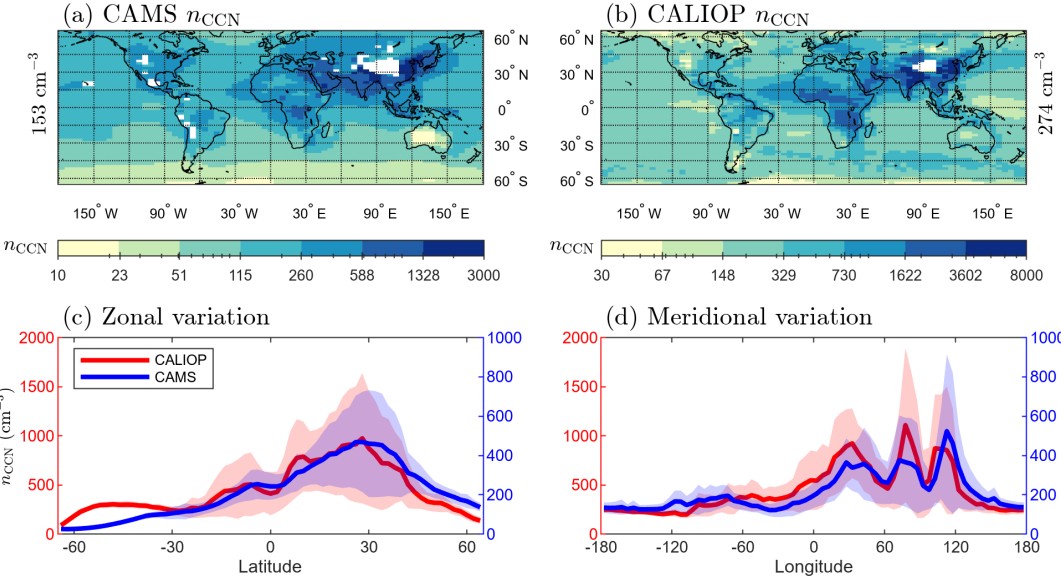

**Figure 1.** Global climatology of cloud condensation nuclei (CCN) concentration ($n_{CCN}$) at altitudes below 2 km. (a) Global climatology derived from CAMS reanalysis. (b) Global climatology derived from CALIOP spaceborne lidar. Median $n_{CCN}$ values are displayed on the lateral edges of panels (a) and (b). (c) Zonal variations of $n_{CCN}$ climatology. (d) Meridional variations of $n_{CCN}$ climatology. The semi-transparent patch in (c) and (d) represents one standard deviation. Note the different color scales in top row, and the varying right and left y-axis limits in bottom row. CALIOP and CAMS data from June 2006 through December 2021 are used to generate the climatology.

bridges the gap between the global datasets, providing insights for their future application and development. Ultimately, this work aims to establish a benchmark for applying and developing CCN-retrieval algorithms in the context of aerosol-cloud interactions.

## 2 Results

### 2.1 $n_{CCN}$ climatology in CAMS and CALIOP

We first compare the spatial variations in $n_{CCN}$ climatology at a supersaturation of 0.20 % for altitudes relevant to liquid clouds (< 2 km) in CALIOP and CAMS datasets (Fig. 1). CAMS $n_{CCN}$ ranges primarily between 28 cm$^{-3}$ and 619 cm$^{-3}$ (5$^{th}$ and 95$^{th}$ percentiles), with a global median of 153 cm$^{-3}$ (Fig. 1a). In contrast, CALIOP retrievals exhibit a broader range, varying from 107 cm$^{-3}$ to 1445 cm$^{-3}$, with a global median of 274 cm$^{-3}$ (Fig. 1b). Overall, CALIOP-derived $n_{CCN}$ are approximately 79 % higher than those from CAMS. This difference is also reflected in the magnitudes of their zonal and meridional variations (Fig. 1c and 1d). Despite the discrepancies in magnitudes, the zonal and meridional patterns in both datasets are quite similar, with identical peaks and troughs across most regions except in the Southern Hemisphere (SH). The difference in the SH primarily originates from the retrievals over oceans, where CALIOP-derived concentrations are significantly higher than those from

CAMS (by 208 %). This difference is particularly large for latitudes south of 45° S, where the median CAMS $n_{\mathrm{CCN}}$ (33 cm$^{-3}$) is roughly seven times lower than that from CALIOP (263 cm$^{-3}$).

Both datasets show higher $n_{\mathrm{CCN}}$ in the Northern Hemisphere (NH) compared to SH. However, this contrast is significantly stronger in CAMS (160 %) compared to CALIOP (20 %). This hemispheric difference in CAMS is particularly pronounced over oceans (121 %) compared to land (59%) and far exceeds the contrast observed in CALIOP (18 % over land and 10 % over oceans). Interestingly, the hemispheric contrast persists in CAMS even over pristine oceans far from continental influence, where CALIOP exhibits homogeneous concentrations. Heterogeneity in CALIOP's oceanic $n_{\mathrm{CCN}}$ is primarily confined to transatlantic dust transport in the tropics and the extra-tropical SH region of strong westerly winds. Since dust is not considered CCN-active in CAMS, the $n_{\mathrm{CCN}}$ peak over the tropical Atlantic Ocean observed in CALIOP is less pronounced in CAMS. Furthermore, the CCN belt in the Southern Ocean (SO), though visible particularly in sea-salt $n_{\mathrm{CCN}}$ in CAMS (see Supplementary Fig. S1), does not appear in the total $n_{\mathrm{CCN}}$ climatology due to low sea-salt concentrations. When comparing the contrast between land and ocean $n_{\mathrm{CCN}}$, we find similar values for CAMS and CALIOP in the NH, with land values 65 % and 86 % higher than those over oceans, respectively. However, this difference in the SH is more pronounced in CAMS (130 %) than in CALIOP (73 %) due to substantially lower concentrations in CAMS over SH oceans. Refer to Table A1 for the median values used in these calculations.

### 2.1.1 Regional consistency with in-situ observations

To evaluate the datasets, we compare the $n_{\mathrm{CCN}}$ climatology from the global datasets with in-situ observations (from the literature, refer Table A1) for 16 regional domains encompassing major continents and ocean basins (geographical boundaries provided in Fig. 2a). Among all, Asia exhibits the highest overall $n_{\mathrm{CCN}}$ (Fig. 2b), within which Southeast Asia shows the highest concentration, followed by South Asia and West Asia, consistently across CAMS, CALIOP, and in-situ retrievals. Other continental and oceanic domains follow in decreasing order. Both datasets indicate cleaner SH oceanic regions (Southeast Pacific, South Atlantic, Indian Ocean, and Southern Ocean) compared to the NH oceans (Northeast Pacific and North Atlantic). However, this hemispheric order is opposite in the in-situ measurements, where concentrations in the SH Atlantic and Pacific oceans exceed their respective NH counterparts. It is important to consider that while the regional domains over oceans in this study extend tens of degrees of longitude away from the coast, in-situ observations for ocean environments may be limited in space (close to the coast) and time. For instance, the observations over the Southern Ocean (Humphries et al., 2023) are mostly obtained during the austral summer.

When comparing the magnitudes of $n_{\mathrm{CCN}}$, we observe that CALIOP-derived concentrations are consistently higher than those of CAMS across all regions except North America. These elevated values in the CALIOP data are expected because the retrieval in CALIOP assumes a fixed CCN-activation radius, above which all aerosols are considered CCN-active regardless of their hygroscopicity. This assumption can lead to overestimation of $n_{\mathrm{CCN}}$ in urban continental regions (Southeast and South Asia, and Southern Africa) influenced by black carbon and regions downwind. CAMS, on the other hand, considers 80 % of black carbon aerosols to be hydrophobic (and thus not contributing to $n_{\mathrm{CCN}}$) (Block et al., 2024). Additionally, CAMS excludes dust as a potential CCN source, which is accounted for in CALIOP. These differences in the assumptions in

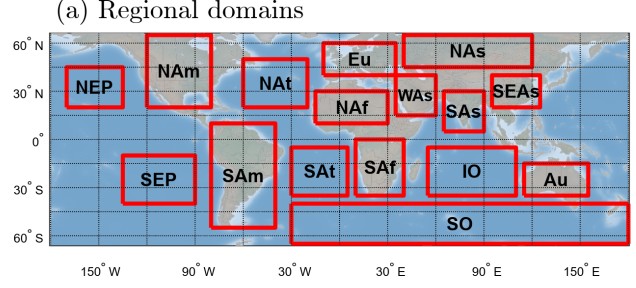

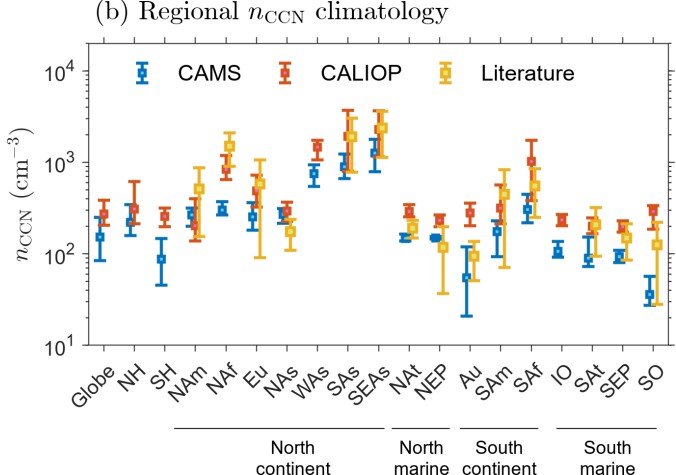

**Figure 2.** Comparison of regional cloud condensation nuclei (CCN) concentrations $n_{\text{CCN}}$ with in-situ measurements. (a) Geographical extent of regional domains considered in this study. (b) Comparison of median $n_{\text{CCN}}$ for various domains derived from CAMS reanalysis (blue), CALIOP (red), and in-situ observations from literature (yellow). Error bars for CAMS and CALIOP represent the geographic interquartile range of $n_{\text{CCN}}$. Error bars for in-situ observations represent the temporal $n_{\text{CCN}}$ variations at the specific measurement locations (refer to Table A1). CALIOP and CAMS data from June 2006 through December 2021 are used to produce the regional climatology. NH: Northern Hemisphere; SH: Southern Hemisphere; NAm: North America; NAf: Northern Africa; Eu: Europe; NAs: North Asia; WAs: West Asia; SAs: Southern Asia; SEAs: Southeast Asia; NAt: North Atlantic; NEP: Northeast Pacific; Au: Australia; SAm: South America; SAf: South Africa; IO: Indian Ocean; SAt: South Atlantic; SEP: Southeast Pacific.

CALIOP and CAMS in terms of aerosol hygroscopicity, activation size, and CCN activity may naturally lead to lead to higher concentrations in CALIOP compared to CAMS. Other factors may also contribute to these differences. For example, CALIOP's aerosol extinction coefficient may not correlate well with $n_{\text{CCN}}$ in complex aerosol mixtures with varying hygroscopicity (Choudhury and Tesche, 2022a). Additionally, inaccuracies in the representation of aerosol sources and sinks in CAMS may bias the derived $n_{\text{CCN}}$ (Moore et al., 2013). More details on the inherent differences between the global datasets are discussed in Section A1. Despite these discrepancies, this regional comparison with in-situ measurements suggests that the global datasets adequately capture the observed variations in $n_{\text{CCN}}$ climatology for most regions. CALIOP appears to represent the upper

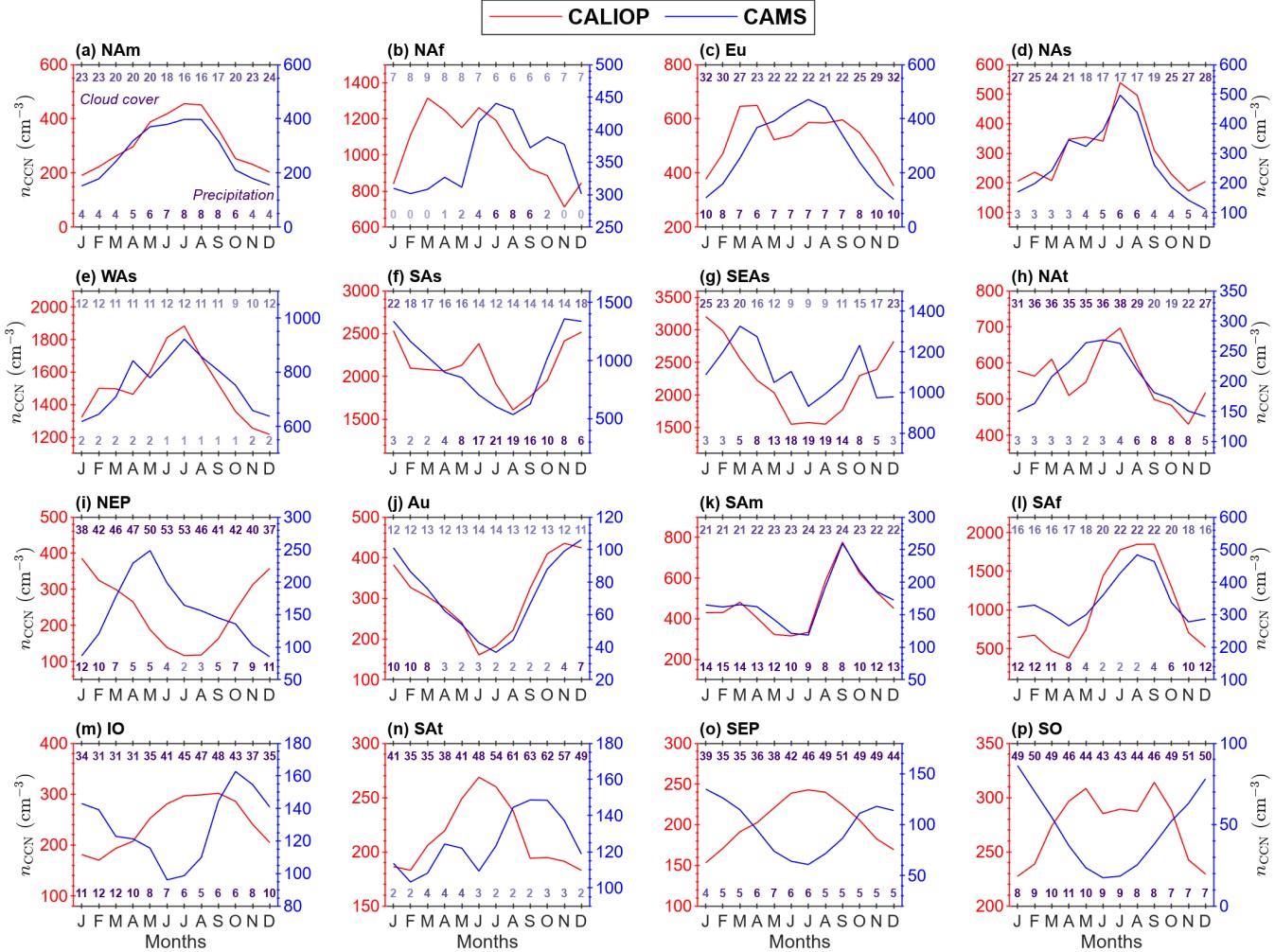

**Figure 3.** Monthly variations in cloud condensation nuclei concentrations ($n_{\text{CCN}}$) for various regions. Red lines represent $n_{\text{CCN}}$ derived from spaceborne CALIOP, and blue lines represent $n_{\text{CCN}}$ from CAMS reanalysis. Panels (a) to (i) correspond to Northern Hemisphere regions, while panels (j) to (p) represent Southern Hemisphere regions. Note the separate y-axes for CALIOP (left) and CAMS (right). The numbers at the top and bottom of each panel represent the monthly climatology of low cloud cover (in %) from CERES and precipitation (in cm) from GPCP product, respectively, with the opacity of the numbers proportional to their magnitude. Datasets from June 2006 through December 2021 are used to generate the monthly climatology.

bound, while CAMS represents the lower bound of $n_{\text{CCN}}$, highlighting their potential for constraining $n_{\text{CCN}}$ even in regions lacking in-situ measurements.

## 2.2 Monthly $n_{\text{CCN}}$ variations

To understand how well the datasets capture the seasonal $n_{\text{CCN}}$ cycles, we analyze the average monthly variations in $n_{\text{CCN}}$ derived from CALIOP and CAMS for different regional domains (see Fig. 3). Both datasets exhibit a consistent pattern for most continental regions, with $n_{\text{CCN}}$ peaking in summer (boreal in NH and austral in SH) and reaching a minimum in winter. This pattern aligns with regional precipitation cycles (shown at the bottom of all panels of Fig. 3), where wet winters lead to precipitation scavenging of airborne particles, resulting in lower $n_{\text{CCN}}$ compared to dry summers. Exceptions include the monsoon-influenced South and Southeast Asia regions, which experience a summer minimum and winter maximum in $n_{\text{CCN}}$ due to prolonged summer rainfall. Both datasets adequately capture this seasonal pattern driven by the monsoon cycle.

However, the datasets show contrasting variations for all oceanic regions, except for the North Atlantic region. CALIOP exhibits a summer minimum and winter maximum in oceanic $n_{\text{CCN}}$, while CAMS generally shows a spring–summer maximum and winter minimum. The variations in CALIOP align with the seasonal cycle of near-surface wind speeds over oceans (Yu et al., 2020). Higher wind speeds increase sea spray aerosol concentrations in marine environments by enhancing wave breaking and bubble bursting (Revell et al., 2019; Humphries et al., 2023), which may contribute to the observed CCN cycles in CALIOP. However, oceanic $n_{\text{CCN}}$ are also influenced by factors beyond sea spray aerosols, such as biogenic emissions, which follow a seasonal pattern of summer maximum and winter minimum (Lana et al., 2011; Revell et al., 2019), more in line with CAMS. Studies in pristine oceans have shown that while sea salt aerosols primarily contribute to aerosol mass, sulphates from biogenic emissions dominate particle or CCN concentrations (Ayers and Gras, 1991; Gras and Keywood, 2017; Humphries et al., 2023). Consequently, in-situ-derived $n_{\text{CCN}}$ variations in these regions closely follow biogenic emission patterns (Gras, 1990; Ayers and Gras, 1991; Gras and Keywood, 2017), exhibiting a spring–summer maximum and winter minimum. As a result, cloud droplet number concentrations ($N_{\text{d}}$), a parameter sensitive to changes in $n_{\text{CCN}}$, also displays a spring–summer maxima and winter minima in pristine Southern Oceans (McCoy et al. (2015); Mace and Avey (2017); see also Fig. S2 in the supplementary material). These seasonal CCN cycles are well represented in CAMS but not in CALIOP. Additionally, the austral summer concentrations in CAMS for the Southern Ocean (Fig. 3p) are comparable to the in-situ observations reported by Humphries et al. (2023), which were mostly obtained during the austral summer. This observation contrasts with the results inferred from climatological concentrations in Fig. 2, where CALIOP misleadingly appears to show better agreement.

Further investigation reveals that while the total $n_{\text{CCN}}$ seasonal cycles in most oceanic regions are opposite in CALIOP and CAMS, the marine $n_{\text{CCN}}$ in CALIOP aligns closely with CAMS's sea salt $n_{\text{CCN}}$, with both exhibiting a summer minimum and winter maximum (first and third columns in Fig. 4). This similarity can be attributed to the similar seasonal cycles of CALIOP's marine extinction coefficients ($\alpha_{\text{M}}$; second column in Fig. 4) and CAMS's sea salt mass mixing ratio (MMR$_{\text{SS}}$; fourth column in Fig. 4), the primary parameters from which their respective $n_{\text{CCN}}$ are calculated (Choudhury and Tesche, 2022a; Block et al., 2024). Since aerosol mass in pristine oceans consists primarily of coarse mode sea salt particles (Humphries et al. (2023); fourth column in Fig. A1), $\alpha_{\text{M}}$ is expected to be proportional to MMR$_{\text{SS}}$, as these coarse particles dominate light scattering. $n_{\text{CCN}}$ in CALIOP's retrieval algorithm is proportional to aerosol extinction coefficient (Shinozuka et al., 2015; Choudhury and Tesche, 2022a), so the seasonal $n_{\text{CCN}}$ cycles in CALIOP for pristine oceans follow the variations in sea salt

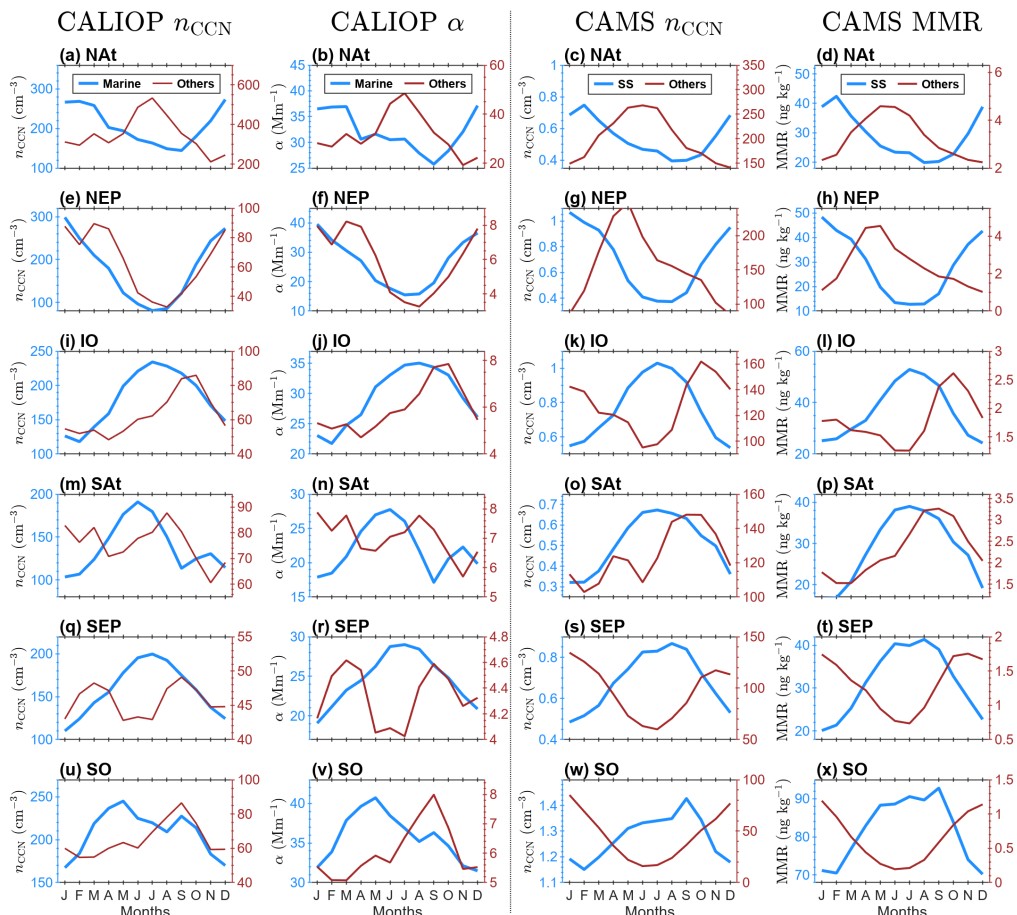

**Figure 4.** Monthly variations in cloud condensation nuclei concentrations ($n_{CCN}$), extinction coefficients ($\alpha$) and mass mixing ratios (MMR) for six oceanic domains. Blue lines represent marine aerosols in CALIOP and sea-salt aerosols in CAMS, while brown line represent contributions from other aerosol species. Panels in the first and second column depict the marine and non-marine $n_{CCN}$ and $\alpha$ derived from CALIOP, respectively. The third and fourth column show the sea-salt and non-sea-salt $n_{CCN}$ and MMR derived from CAMS, respectively. Datasets from June 2006 through December 2021 are used to generate the monthly climatology.

aerosols. Given that sulphates are the primary contributors to $n_{CCN}$ in these regions (Ayers and Gras, 1991; Gras and Keywood, 2017; Humphries et al., 2023), the separation of marine extinction coefficients in CALIOP into contributions from sea salt and biogenic aerosols is crucial for accurately representing $n_{CCN}$ cycles over pristine oceans. This separation, however, requires precise quantification of their lidar ratios and depolarization properties (Tesche et al., 2009), which is currently lacking. On the other hand, CAMS, which can distinguish between different oceanic aerosol species such as sulphates, hydrophilic organic matter, and sea salt, better captures the overall $n_{CCN}$ variations in pristine marine environments.

Nevertheless, CAMS may significantly underestimate the contribution of sea salt aerosols to oceanic $n_{CCN}$ (third column of Fig. 4 and Fig. A1), which can be as high as 8–51 % of the total $n_{CCN}$ and may increase to 100 % at higher surface wind

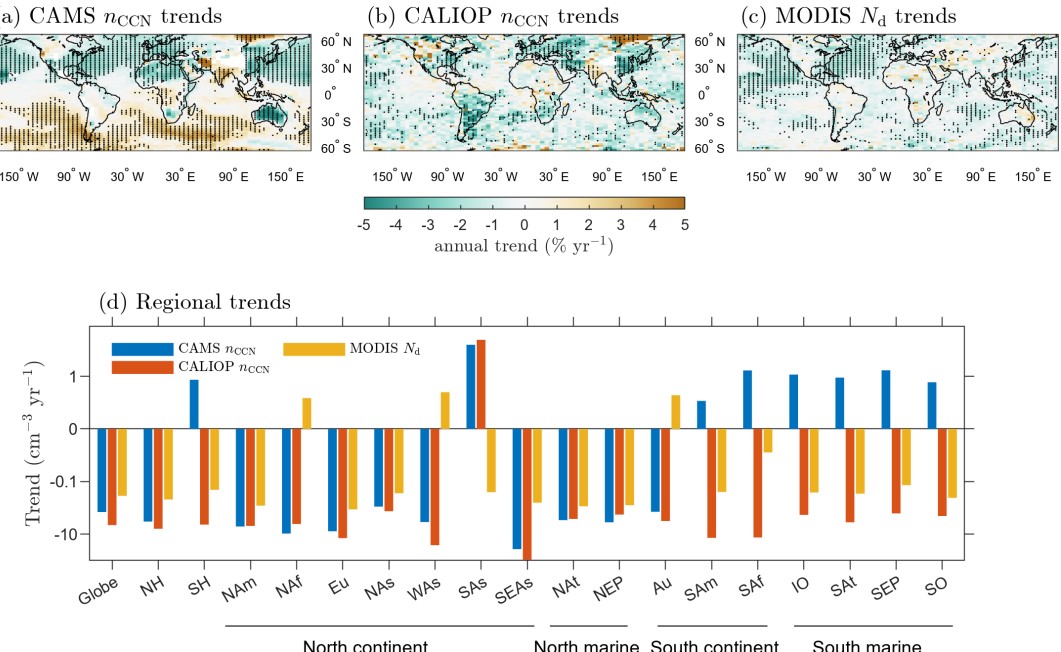

**Figure 5.** Comparison of global and regional trends computed using annual time series. (a) Trends in cloud condensation nuclei concentrations ($n_{CCN}$) derived from CAMS reanalysis. (b) Trends in $n_{CCN}$ from CALIOP. (c) Trends in cloud droplet number concentrations ($N_d$) derived from MODIS. (d) Regional trends in $n_{CCN}$ derived from CAMS reanalysis (blue), $n_{CCN}$ from CALIOP (red), and $N_d$ from MODIS (yellow) are compared. Dots in panels (a)–(c) indicate the grids where the trend is statistically significant. The absolute values of the trends in panels (a)–(c) are shown in supplementary Fig. S3. Trends in $n_{CCN}$ from CALIOP and CAMS are produced using data from 2007 to 2021, while data from 2007 to 2020 are used for $N_d$. Annual time series for the regional domains are provided in Fig. S4 in the supplementary materials.

speeds (Fossum et al., 2018). This underestimation could stem from an underrepresentation of small-mode sea-salt aerosol mass in CAMS (see fourth column of Fig. A1). Another plausible reason may be the size distribution assumed in CAMS's CCN-retrieval algorithm, which may not accurately represent small-mode sea-salt aerosols. Such factors likely contribute to
the observed low $n_{CCN}$ values in CAMS compared to in-situ observations for SH oceanic domains (see Fig. 2b). Additionally, the inaccurate representation of CCN generated from new particle formation processes (McCoy et al., 2021; Mace et al., 2023, 2024) may further contribute to the underestimation of CCN in CAMS. However, due to limited in-situ observations across different regions in the SH oceans, the contribution of these aerosol species to total oceanic $n_{CCN}$, as well as their seasonal variations across different oceanic regions, remains uncertain. It is important to note that SH oceans are the primary
contributor to global low-level cloud cover (see top of all panels in Fig. 3, and Fig. A2b). These inconsistencies observed in the global $n_{CCN}$ datasets in such cloud-rich regions demand further improvements in the underlying CALIOP and CAMS datasets, as well as in the associated CCN-retrieval algorithms, to better constrain aerosol-cloud interactions.

## 2.3 Reconciling trends in $n_{CCN}$ and $N_d$

Quantifying trends in $n_{CCN}$ is crucial for comprehending the present dynamics of radiative forcing due to ACIs and for projecting future changes. Recent decades have witnessed declining aerosol emission rates and aerosol loadings over land (Collaud Coen et al., 2020; Quaas et al., 2022) and oceans (IMO, 2019; Gryspeerdt et al., 2019) due to stricter emission policies. An exception is the South Asia region, where aerosol emissions have been increasing in the 21$^{st}$ century (Jin et al., 2023). These emission trends are also expected to be reflected in $N_d$ because of their strong sensitivity to changes in $n_{CCN}$ (McCoy et al., 2018; Quaas et al., 2022). Therefore, we expect the annual trends in $n_{CCN}$ and $N_d$ to be similar to the emission trends.

Over NH regions, the emission trends are reflected in both the $n_{CCN}$ datasets (Figs. 5a and 5b). As expected, all regions except South Asia exhibit a declining $n_{CCN}$ trend (see Fig. 5c and Table A1). The trends in $N_d$ are also consistent with those in $n_{CCN}$ from both CALIOP and CAMS (Fig. 5c), with exceptions only observed over dust-influenced regions (Northern Africa and West Asia). This discrepancy may be attributed to the hydrophobic nature of fresh mineral dust, which may not readily act as CCN due to a lack of mixing or coating with water-soluble aerosols (Garimella et al., 2014).

Over SH regions, CALIOP shows declining $n_{CCN}$ trends across all domains. $N_d$ trends are mostly negative as well consistent with CALIOP, except for dust-influenced Australia (Au) domain. Of particular interest are the spatially uniform and statistically significant increasing trends in CAMS-derived $n_{CCN}$ at altitudes below 2 km over most SH oceanic regions. This finding not only contradicts the negative trend observed in $N_d$ and CALIOP-derived $n_{CCN}$ but also the expected decreasing trend inferred from previous ship emission reports (Quaas et al., 2022). The trend even exists in the mass mixing ratios in CAMS data (see Fig. S11 in the supplementary materials), particularly corresponding to sulphate aerosol species. It is worth noting that the increasing SH $n_{CCN}$ trends in CAMS coincide with trends in AOD derived from MODIS (see Fig. A3 in the supplementary material). Since MODIS AOD is used to constrain the CAMS aerosol reanalysis (Inness et al., 2019a), a proportionality between AOD and CAMS-derived $n_{CCN}$ is inherent in homogeneous marine environments (Block et al., 2024), and may contribute to the observed increasing trends in CAMS. These inconsistencies over pristine oceans, where the trends in aerosol loadings differ between different spaceborne retrievals (Quaas et al., 2022), question the representativeness of the $n_{CCN}$ and $N_d$ retrievals, making it challenging to derive their inter-relationship, a parameter key to quantifying ACIs.

## 3 Conclusions

The closure study presented here shows good consistency between the independent CALIOP and CAMS global $n_{CCN}$ datasets in continental environments. However, significant discrepancies emerge over most pristine oceans, not only in $n_{CCN}$ climatology but also in their monthly and annual variations. While the seasonal cycles of oceanic $n_{CCN}$ derived from CAMS largely align with previous in-situ observations (Gras, 1990; Ayers and Gras, 1991; Gras and Keywood, 2017; Humphries et al., 2023) and the variations in $N_d$, CAMS likely underestimates the contributions from sea salt and secondary biogenic $n_{CCN}$. In contrast, the seasonal $n_{CCN}$ cycles in CALIOP are not representative, likely due to its inability to resolve marine $n_{CCN}$ into sea salt and sulphate (from biogenic emission) components.

The results, however, are completely opposite for annual trends in $n_{CCN}$ and $N_d$. While trends in CAMS and CALIOP generally agree across most NH regions, they diverge significantly in the SH. CALIOP consistently shows a declining $n_{CCN}$ trend in these regions, which aligns with previous reports (IMO, 2019; Gryspeerdt et al., 2019; Quaas et al., 2022) and the decreasing trend in $N_d$, while CAMS exhibits an anomalous increasing $n_{CCN}$ trend over SH oceans. This geographically limited disagreement, restricted to pristine oceans with limited in-situ measurements, raises questions about the adequacy of aerosol inventories used by CAMS in SH oceans, a known issue in climate models (Moore et al., 2013). These discrepancies in cloud-rich pristine oceans are particularly concerning because cloud properties in these regions are highly sensitive to even small perturbations in aerosol concentrations (Moore et al., 2013; Gryspeerdt et al., 2023).

Caution should therefore be taken when using these $n_{CCN}$ datasets in the pristine oceans of SH. Future research efforts should focus on first separating the sea salt and biogenic components of marine aerosols in CALIOP, and second, on accurately quantifying the sources and sinks of CCN and their long-term cycles in remote SH oceans for improving the representativeness of aerosol inventories in CAMS. An alternative approach could involve the further development of advanced data-driven techniques to derive global CCN dataset (Redemann and Gao, 2024). These efforts are crucial to refine the global $n_{CCN}$ datasets and ultimately to reduce the uncertainties in $ERF_{ACI}$. In conclusion, the aerosol-limited environments of SH oceans are identified as a significant source of uncertainty in the present effort to quantify a highly resolved global $n_{CCN}$ dataset.

*Data availability.* All datasets used in this work are opensource. The CALIPSO Level 2 Aerosol Profile product can be downloaded from https://doi.org/10.5067/CALIOP/ (NASA/LARC/SD/ASDC, 2018). CALIOP CCN data can be accessed at https://doi.pangaea.de/10.1594/ PANGAEA.956215 (last access: December 25, 2024; Choudhury and Tesche, 2023b). CAMS mass mixing ratios were acquired from the Copernicus Atmosphere Monitoring Service (CAMS) Atmosphere Data Store (ADS) https://ads.atmosphere.copernicus.eu/datasets/ cams-global-reanalysis-eac4-monthly?tab=overview (last access: December 25, 2024; Inness et al., 2019b). CAMS-derived CCN data can be downloaded from https://doi.org/10.26050/WDCC/QUAERERE_CCNCAMS_v1 (last access: December 25, 2024; Block, 2023). CERES SYN level 3 product were obtained from the NASA Langley Research Center Atmospheric Science Data Center and can be accessed at https://ceres-tool.larc.nasa.gov/data (last access: December 25, 2024). MODIS-derived cloud droplet number concentrations can be downloaded from https://dx.doi.org/10.5285/864a46cc65054008857ee5bb772a2a2b (last access: December 25, 2024; Gryspeerdt et al., 2022). MODIS Aqua aerosol product (last access: December 25, 2024; Platnick et al., 2017a) are obtained from the Level-1 and Atmosphere Archive and Distribution System (LAADS) Distributed Active Archive Center (DAAC), located in the Goddard Space Flight Center in Greenbelt, Maryland (https://ladsweb.nascom.nasa.gov/). Precipitation data are obtained from the Global Precipitation Climatology Project (GPCP) Monthly Analysis Product data provided by the NOAA PSL, Boulder, Colorado, USA, from their website at https://psl.noaa.gov (last access: December 25, 2024).

## Appendix A: Methods

### A1  Global $n_{\mathrm{CCN}}$ datasets

CALIOP dataset provides $n_{\mathrm{CCN}}$ at a supersaturation of 0.20 %. It is available on a uniform latitude-longitude grid of resolution 2º by 5º, a vertical grid resolution of 60 m extending from mean sea level to a height of 8 km above mean sea level, and a temporal resolution of one month. The dataset is derived from more than 15 years of CALIOP level 2 aerosol profile product from June 2006 to December 2021 (NASA/LARC/SD/ASDC, 2018). It is based on a CCN-retrieval algorithm (Choudhury and Tesche, 2022a) that integrates the CALIOP-derived height-resolved information on the aerosol-type-specific extinction coefficient and microphysical properties from CALIOP's aerosol model with the optical modelling capabilities of the MOPSMAP (Modelled Optical Properties of enseMbles of Aerosol Particles; Gasteiger and Wiegner, 2018) package. Essentially, the algorithm adjusts the normalized size distributions within the aerosol model to match the extinction coefficient. These adjusted size distributions are then used to estimate particle number concentrations relevant for CCN activation. Aerosol-type-specific CCN parameterizations are then applied to calculate $n_{\mathrm{CCN}}$ at a supersaturation of 0.20 % for continental (comprising of clean, polluted, and smoke aerosols), dust, and marine aerosols. The algorithm accounts for hygroscopic growth of hydrophilic aerosols (continental and marine aerosols) under humid conditions using the $\kappa$-parameterization within MOPSMAP package. Evaluations of the algorithm have demonstrated good agreement with independent ground-based and airborne in-situ measurements across diverse geographic locations, with a combined normalized mean bias of $\approx 22$ % and a normalized absolute error of $\approx 61$ % (Choudhury et al., 2022; Choudhury and Tesche, 2022b; Aravindhavel et al., 2023; Choudhury and Tesche, 2023a). The resulting CALIOP-derived $n_{\mathrm{CCN}}$ has also been utilized in quantifying the CCN activation ratio for liquid clouds (Alexandri et al., 2024).

CAMS $n_{\mathrm{CCN}}$ dataset (Block et al., 2024) is derived from CAMS aerosol reanalysis of mass mixing ratios (Inness et al., 2019b) and provides $n_{\mathrm{CCN}}$ at supersaturations ranging from 0.1 % to 1 %. The $n_{\mathrm{CCN}}$ dataset retains the native resolution of CAMS reanalysis data and is available on a uniform horizontal grid of resolution 0.75º by 0.75º and a vertical grid with 60 hybrid sigma–pressure levels extending from the surface to 0.1 hPa. The CCN-retrieval algorithm in CAMS utilizes a box-model framework (O'Connor et al., 2014; West et al., 2014) to convert the mass mixing ratios of five aerosols species—sulfate, mineral dust, black carbon (hydrophobic and hydrophilic), organic matter (hydrophobic and hydrophilic), and sea salt—into total number concentrations. Subsequently, these concentrations are combined with normalized size distributions derived from the aerosol module of the European Centre for Medium-Range Weather Forecasts (ECMWF) Integrated Forecasting System (IFS) model (Benedetti et al., 2009) to estimate the actual aerosol size distribution. The size distributions of hydrophillic aerosols are then coupled with auxillary meteorological parameters and used in modified Kappa-Köhler theory (Pöhlker et al., 2023) to calculate the activated $n_{\mathrm{CCN}}$ at various supersaturations. Consistent with the CAMS model's assumption of completely hydrophobic dust with no consideration of internal mixing or external coating mechanisms, dust is excluded in the CCN calculations. Initial validation results using surface in-situ CCN observations at continental and coastal Atmospheric Radiation Measurement (ARM) network sites have shown promising results, with an acceptable bias factor of 1.29 (Block et al., 2024).

### A1.1 Limitations of $n_{\mathrm{CCN}}$ datasets

CALIOP $n_{\mathrm{CCN}}$ dataset is subject to uncertainties arising from errors in the underlying CALIOP products and approximations within the CCN-retrieval algorithm. Uncertainties in CALIOP extinction coefficients can reach 30 %. Assuming fixed aerosol-type-specific size distributions introduces additional uncertainty, estimated to be a factor of 1.5–2 (Choudhury and Tesche, 2022a). Further, the algorithm assumes an aerosol-species dependent fixed CCN activation radius (50 nm for continental and marine aerosols, and 100 nm for mineral dust at a supersaturation of 0.20 %). Using a fixed CCN activation size (assuming all larger particles are CCN active) may result in about a 20 % overestimation in the final CCN product (Choudhury and Tesche, 2022b). Accounting for all these limitations, the overall uncertainty associated with the CALIOP-derived CCN dataset is expected to be a factor of 2–3 (Choudhury and Tesche, 2023a). Moreover, the CALIOP dataset is produced using only cloud-free aerosol profiles. This can lead to sampling bias in regions with significant cloud cover, potentially leading to the differences observed between the CALIOP and CAMS datasets. However, there appears to be no clear relationship between the correlation of the CALIOP and CAMS $n_{\mathrm{CCN}}$ datasets and the sampling frequency of CALIOP (Fig. A4).

Similarly, uncertainties in CAMS $n_{\mathrm{CCN}}$ dataset may stem from the source CAMS aerosol reanalysis product and the CCN-estimation methodology. CAMS aerosol product is constrained by satellite-derived AOD retrievals, particularly the MODIS dark target and deep blue AOD retrievals at 0.55 um (Platnick et al., 2017b) and Advanced Along-Track Scanning Radiometer (AATSR) retrieved AOD (Popp et al., 2016). Therefore, uncertainties in AOD retrievals can propagate into the CAMS reanalysis and ultimately the $n_{\mathrm{CCN}}$ product. Additionally, missing aerosol sources in the CAMS emission inventory (Moore et al., 2013; Errera et al., 2021) can introduce uncertainties, especially in remote areas with sparse observations, limiting the effectiveness of emission parameterizations implemented in the aerosol model. Furthermore, unlike the approach in CALIOP, the CAMS-based retrieval excludes mineral dust. Studies have demonstrated that mineral dust may be a potential CCN source, particularly when coated or internally mixed with water-soluble hydrophilic aerosols (Kumar et al., 2009; Bègue et al., 2015). This exclusion may thus lead to an underestimation in the final $n_{\mathrm{CCN}}$ product.

### A2 Spaceborne cloud and precipitation data

$N_{\mathrm{d}}$ data for low-level liquid clouds are derived from the Moderate Resolution Imaging Spectroradiometer (MODIS) aboard Aqua polar orbiting satellite (Gryspeerdt et al., 2022). The dataset is available at a uniform spatial resolution of 1º by 1º with daily temporal resolution spanning from July 2002 and 2020. Low-level cloud cover data are obtained from the Clouds and the Earth's Radiant Energy System (CERES) SYN Edition 4A monthly product (Doelling et al., 2013). This product merges retrievals from CERES, MODIS, and geostationary sensors to construct a global gridded dataset suitable for studying aerosol-cloud interactions. The dataset is operationally available at a latitude-longitude resolution of 1º by 1º starting from July 2002.

Precipitation data are derived from the Global Precpitation Climatology Project (GPCP) monthly product (Adler et al., 2003). This product integrates rainfall data obtained from several platforms, including satellites, in-situ soundings, and rain gauges, to generate a global monthly precipitation dataset on a uniform horizontal resolution of 2.5 º available from 1979.

## A3 Data harmonization, trend estimation, and averaging methodologies

CCN, cloud, and precipitation parameters are considered between latitudes of 65º N and 65ºS. Data at higher latitudes are not considered due to the uncertainties associated with MODIS observations at high solar zenith angles (Grosvenor and Wood, 2014; Grosvenor et al., 2018) and the lack of validation for CALIOP retrievals at these latitudes. Horizontal grids of all datasets are harmonized by transforming them to the coarser 2º by 5º latitude-longitude grid of CALIOP using bilinear interpolation. We exclude CAMS data in grids surrounding Mauna Loa and Altzomoni due to documented biases in CAMS aerosol emission datasets over these regions (Inness et al., 2019a).

To specifically focus on the liquid clouds, which are most relevant for aerosol-cloud interactions, average $n_{\mathrm{CCN}}$ between altitudes of 0–2 km are considered in this study. Additionally, a supersaturation of 0.20 % is selected because this value represents a characteristic supersaturation near the base of liquid clouds. Temporal averages of CALIOP data are weighted by the number of valid aerosol retrievals within each grid cell (Choudhury and Tesche, 2023a). Horizontal averages in CALIOP and CAMS are weighted by the area of the latitude-longitude grids. Trends in $n_{\mathrm{CCN}}$ and $N_{\mathrm{d}}$ are estimated using the non-parametric Mann-Kendall trend test, as it does not require any assumptions about the distribution of the time series data and is more robust in handling outliers (Mann, 1945; Kendall, 1975). Annual trends computed using linear regression are shown in Fig. S4 in the supplementary material. Monthly and annual statistics are calculated using data between 2007 and 2021 for CALIOP- and CAMS-derived $n_{\mathrm{CCN}}$, and between 2007 and 2020 for MODIS-derived $N_{\mathrm{d}}$.

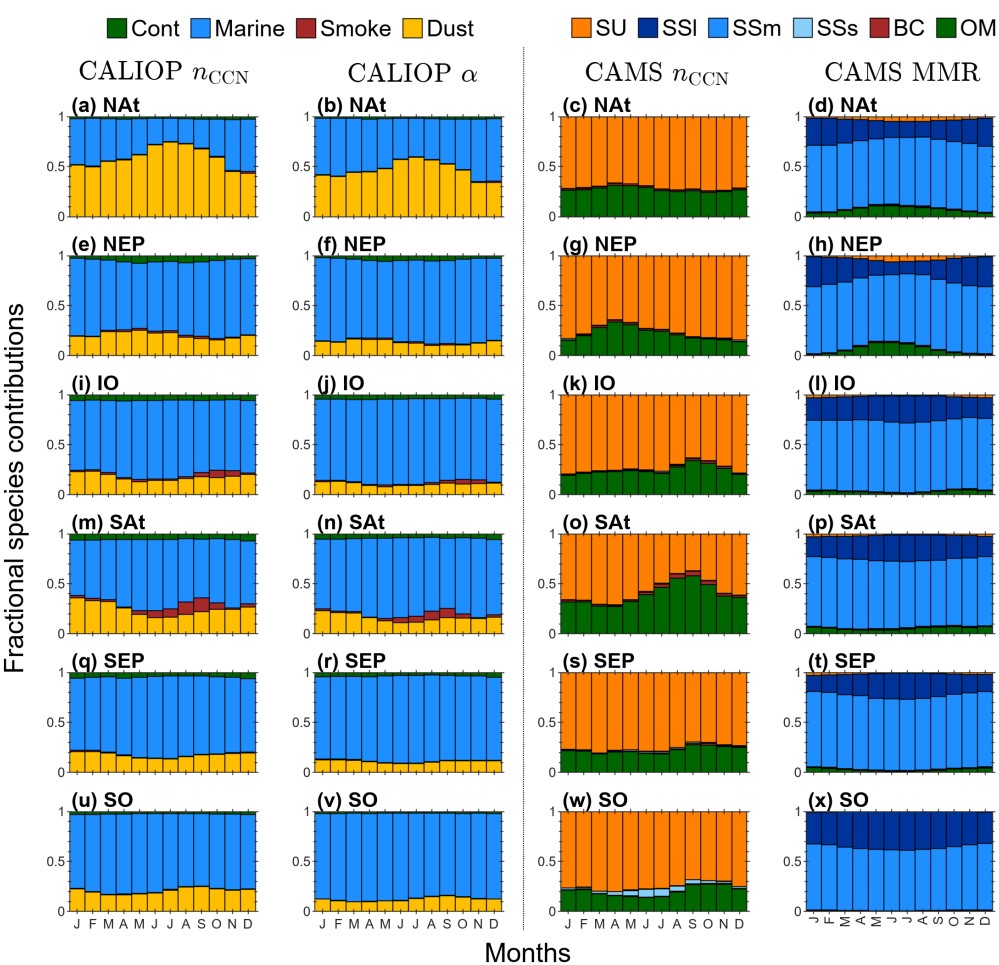

**Figure A1.** Monthly variations in the fractional contributions of different aerosol species to cloud condensation nuclei concentrations ($n_{CCN}$), aerosol extinction coefficients ($\alpha$), and mass mixing ratios (MMR) for six oceanic domains. Panels in the first and second column depict the fractional $n_{CCN}$ and $\alpha$ contributions of different aerosol species (continental, marine, smoke, and dust) in CALIOP. The third and fourth column show the fractional $n_{CCN}$ and MMR contributions of different aerosol species in CAMS (sulphate(SU) , sea salt large (SSl), sea salt medium (SSm), sea salt small (SSs), black carbon (BC), and organic matter (OM)). Datasets from June 2006 to December 2021 are used to generate the monthly climatology. Fractional contributions for other regional domains are given in Figs. S5–S8 in the supplementary material.

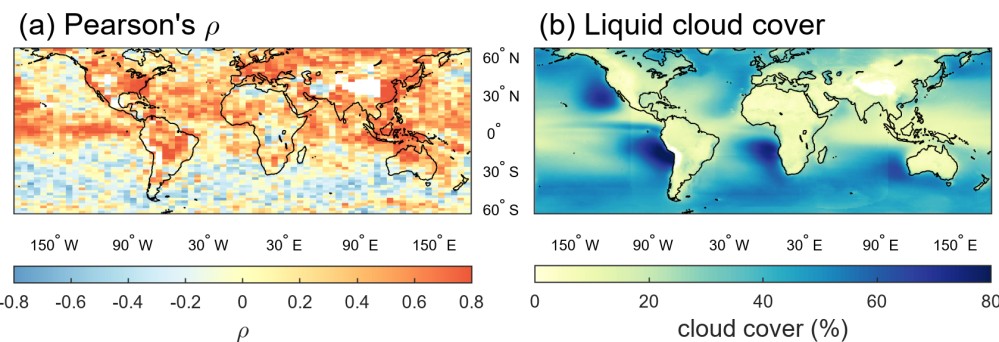

**Figure A2.** Relating correlation between CALIOP and CAMS with global cloud cover. Panel (a): Global map of Pearson's correlation coefficient ($\rho$) between monthly mean cloud condensation nuclei concentration ($n_{\mathrm{CCN}}$) derived from spaceborne CALIOP and CAMS reanalysis datasets. Panel (b): Low-level cloud cover climatology (in %) derived from CERES SYN product.

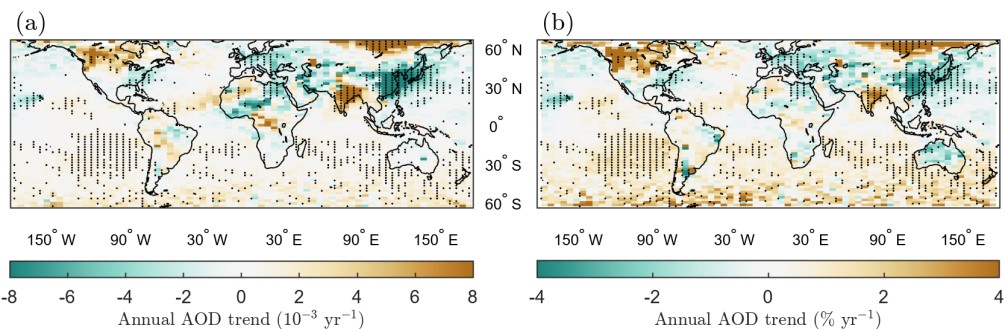

**Figure A3.** Global map of annual trend in MODIS aerosol optical depth (AOD) derived using combined dark target and deep blue algorithms. Panel (a) shows the trend in 10⁻³ yr⁻¹ and panel (b) in % yr⁻¹. Dots in each panel indicate the grids where the trend is statistically significant. Data between 2007 and 2021 are used to estimate the trends.

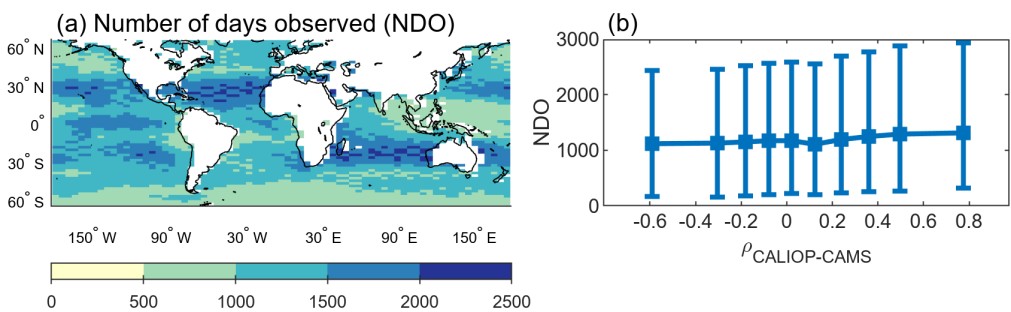

**Figure A4.** Relationship between sampling frequency in CALIOP and correlation between the datasets. (a) Global map of number of days with a valid aerosol retrieval observed by CALIOP within period of June 2006 to December 2021. (b) Median number of valid CALIOP aerosol retrieval over oceans versus Pearson's correlation coefficient between CALIOP and CAMS ($\rho_{\mathrm{CALIOP-CAMS}}$). Error bars denote the interquartile range. Each $\rho_{\mathrm{CALIOP-CAMS}}$ bin consists of 407 data points.

**Table A1.** Median cloud condensation nuclei (CCN) concentration ($n_{CCN}$) at a supersaturation of 0.20 % in cm$^{-3}$ with interquartile range in parentheses, and annual $n_{CCN}$ trend in cm$^{-3}$ yr$^{-1}$ for various regions. Trends in bold indicate statistically significant trends ($p < 0.05$). In-situ $n_{CCN}$ observations and their corresponding references are also provided. Abbreviations are explained in the footnote.

| Region | CALIOP $n_{CCN}$ | Trend CALIOP | CAMS $n_{CCN}$ | Trend CAMS | In situ $n_{CCN}$ | In-situ reference |
|---|---|---|---|---|---|---|
| Globe | 274 (204, 387) | **-4.5** | 153 (84, 250) | **-1.4** | - | - |
| Land | 483 (230, 1071) | **-3.5** | 276 (188, 398) | **-1** | - | - |
| Ocean | 259 (200, 322) | **-1.9** | 130 (71, 183) | **-0.5** | - | - |
| NH | 308 (213, 614) | **-6.2** | 221 (159, 343) | **-3.3** | - | - |
| NH land | 510 (225, 1143) | **-4.4** | 296 (214, 447) | **-1.8** | - | - |
| NH ocean | 275 (212, 395) | **-2** | 179 (147, 267) | **-1.9** | - | - |
| SH | 257 (198, 315) | **-4.3** | 85 (44, 139) | 0.7 | - | - |
| SH land | 432 (245, 831) | **-2.5** | 186 (91, 276) | 0 | - | - |
| SH ocean | 250 (194, 299) | **-2.1** | 81 (41, 124) | **0.8** | - | - |
| NAm | 202 (138, 402) | **-4.9** | 265 (200, 318) | **-5.1** | 515 (154, 876) | Shen et al. (2019) |
| NAf | 837 (648, 1180) | -4.1 | 302 (265, 371) | **-9.5** | 1505 (902, 2108) | Désalmand (1987) |
| Eu | 485 (324, 726) | **-14.2** | 253 (181, 361) | **-7.8** | 578 (91, 1065) | Paramonov et al. (2015) |
| NAs | 293 (226, 367) | -1.3 | 271 (216, 313) | -0.9 | 174 (109, 239) | Asmi et al. (2016) |
| WAs | 1464 (1066, 1734) | **-26.3** | 755 (543, 944) | -3.5 | - | - |
| SAs | 1920 (798, 3713) | **24.4** | 893 (664, 1237) | **15.9** | 1900 (777, 3023) | Jayachandran et al. (2020) |
| SEAs | 2297 (1142, 3649) | **-93.5** | 1256 (790, 1787) | **-37.3** | 2377 (1133, 3023) | Shen et al. (2019) |
| NAt | 291 (252, 346) | -2.6 | 147 (138, 164) | **-2.9** | 191 (149, 233) | Wood et al. (2017) |
| NEP | 231 (197, 265) | -1.8 | 150 (146, 155) | **-3.5** | 117 (37, 197) | Brendecke et al. (2022) |
| Au | 280 (202, 359) | -3.2 | 54 (21, 119) | **-1.4** | 94 (51, 137) | Humphries et al. (2023) |
| SAm | 317 (213, 566) | **-13.9** | 174 (93, 230) | 0.1 | 448 (71, 825) | Shen et al. (2019) |
| SAf | 1017 (379, 1751) | **-13.4** | 306 (218, 445) | 1.7 | 552 (250, 854) | Ross et al. (2003) |
| IO | 236 (203, 268) | **-1.9** | 107 (92, 137) | **1.2** | - | - |
| SAt | 199 (167, 246) | **-3.6** | 90 (73, 152) | **0.9** | 207 (94, 320) | Redemann et al. (2021) |
| SEP | 198 (173, 231) | **-1.6** | 93 (80, 110) | **1.7** | 149 (85, 213) | Allen et al. (2011) |
| SO | 289 (185, 335) | **-2** | 36 (27, 57) | **0.6** | 125 (76, 174) | Humphries et al. (2023) |

NAm: North America [$20° - 65°$N, $120° - 80°$W]; NAf: Northern Africa [$10° - 30°$N, $15°$W $- 30°$E]; Eu: Europe [$40° - 60°$N, $10°$W $- 35°$E]; NAs: North Asia [$45° - 65°$N, $40° - 120°$E]; WAs: West Asia [$15° - 40°$N, $35° - 60°$E]; SAs: Southern Asia [$5° - 30°$N, $65° - 90°$E]; SEAs: Southeast Asia [$20° - 40°$N, $95° - 125°$E]; NAt: North Atlantic [$10° - 35°$N, $60° - 20°$W]; NEP: Northeast Pacific [$20° - 45°$N, $170° - 135°$W]; Au: Australia [$35° - 15°$S, $115° - 155°$E]; SAm: South America [$55° - 10°$S, $80° - 40°$W]; SAf: South Africa [$35° - 0°$S, $10° - 40°$E]; IO: Indian Ocean [$35° - 5°$S, $55° - 110°$E]; SAt: South Atlantic [$35° - 5°$S, $30°$W $- 5°$E]; SEP: Southeast Pacific [$40° - 10°$S, $135° - 90°$W]; SO: Southern Ocean [$65° - 40°$S, $30°$W $- 180°$E]

*Author contributions.* MT conceptualized the initial research idea. GC processed the datasets, compiled the plots, and drafted the initial manuscript. KB, MH, and JQ assisted with the CAMS dataset. TG assisted with the cloud droplet dataset. All authors contributed to the development of the research methodology periodically and in revising the manuscript.

*Competing interests.* At least one of the (co-)authors is a member of the editorial board of Atmospheric Chemistry and Physics.

*Acknowledgements.* The authors would like to acknowledge multiple research funding organizations for supporting this research. GC and TG acknowledge startup funds from Bar-Ilan University. GC was also supported by the German Research Foundation (Deutsche Forschungsgemeinschaft, DFG; grant number 524386224). TG acknowledges funding from the Israel Science Foundation (grant number: 3171/24) and the German Research Foundation (Deutsche Forschungsgemeinschaft, DFG; GZ QU 311/27-1) for the project "CDNC4ACI". MT and GC (initially) were supported by the Franco-German Fellowship Program on Climate, Energy, and Earth System Research (Make Our Planet
Great Again-German Research Initiative (MOPGA-GRI), grant number 57429422) of the German Academic Exchange Service (DAAD), funded by the German Ministry of Education and Research. KB and JQ received funding from the German Federal Ministry for Education and Research (BMBF) project "WarmWorld" (FKZ 01LK2202G). JQ and MH acknowledge funding by the DFG project "VolCloud" (GZ QU 311/23-2). JQ further acknowledges the EU Horizon Europe project CleanCloud (project number 101137639).

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
