# Peer review of "Pristine oceans are a significant source of uncertainty in quantifying global cloud condensation nuclei"

_EGUsphere, 2024_

## Author Comment (AC1)

We are grateful to the editor and reviewers for dedicating their valuable time and effort to reviewing our manuscript. The comments raised by the reviewers have significantly improved the quality and clarity of our work. We agree with most of the feedback provided and have made several changes to the manuscript accordingly. For clarity, we first outline some of the major changes made before proceeding with a point-by-point response to the reviewers' comments. Please note that the reviewers' comments are in black, while our responses are in blue. Any modified or additional text in the revised manuscript is highlighted in red.

**Changes to text:**

Based on the comments of reviewers, we have introduced a new domain in our analysis for the Southern Ocean, defined by latitudes 40S-65S and longitudes 30W-180E. Additionally, we have incorporated comparisons of CCN concentrations from different aerosol species in both CALIOP and CAMS (suggested by Reviewer 2). We compared the aerosol extinction coefficients from CALIOP with the aerosol mass mixing ratios from CAMS, as these are the primary parameters from which CCN concentrations are derived (suggested by Reviewer 2). Incorporating this additional information, along with the several published research suggested by Reviewer 1, has significantly changed our interpretation of the monthly climatology in the datasets. Consequently, Section 2.2 of the manuscript has been substantially revised, and the updated results have been reflected in the Conclusion section of the revised manuscript.

**Changes to figures:**

There have been minor updates to existing figures in the manuscript, such as changes to labels and grid lines (suggested by Reviewer 1). Additionally, the figure layout has been reorganized to conserve space and accommodate the new figures suggested by Reviewer 2. Specifically, Fig. 2 and Fig. A1 from the previous manuscript have been combined into a single figure. Similarly, Fig. 4 and Fig. 5 from the previous version have been merged into Fig. 5 in the revised submission. A new Fig. 4 has been added, showing comparisons between aerosol-species-specific CCN concentrations, extinction coefficients, and mass mixing ratios (as per Reviewer 2). The figures in the Appendix have also been updated. A new Fig. A1, which extends the content of the newly added Fig. 4, has been included. Fig. A3 from the old manuscript has been moved to the supplementary materials, which has led to the re-numbering of Fig. A4 and Fig. A5 from the old version to Fig. A3 and Fig. A4, respectively, in the revised submission. In total, 11 additional figures have been included in the supplementary material to support the revised results and interpretations presented in the manuscript.

**Reviewer 1**

The manuscript by Choudhury et al. (2024) addresses a very important topic examining and comparing two state-of-the-art cloud condensation nuclei (CCN) abundance data sets. One of these is derived from aerosol extinction calculated from the CALIOP lidar data set by Choudhury and Tesche (2023) and the other a blended aerosol model-MODIS aerosol optical depth data set known as Copernicus Atmosphere Monitoring Service (CAMS) aerosol reanalysis (Inness et al., 2019). Using data from roughly 2007-2020, the authors compare the CCN in various regions of the Earth, examine seasonal cycles in these regions using monthly statistics, and finally examine trends over the period of record – also bringing in MODIS derived cloud droplet number concentrations (Nd) in the trend analysis.

The authors find that the CCN data sets present reasonably good agreement in the Northern Hemisphere. However, the agreement is strikingly different in the pristine Southern Hemisphere oceans. This disagreement shows up in the mean statistics with CAM being significantly lower in the annual mean compared to the CALIOP data. These differences extend to the seasonal cycle where the two data sets are largely opposites with the CALIOP data showing a winter maximum and CAM showing a winter minimum. The trends are also different with the CALIOP data showing an overall decreasing trend that is consistent with the MODIS Nd data whereas, over the pristine Southern Ocean, CAM has an increasing trend. The large differences between the Northern and Southern Hemispheres points to structural issues with at least one of the algorithms in regions of low natural AOD. While the authors are careful to present a balanced examination, they do argue that the CALIOP data set is the more reasonable in the regions of disagreement.

Overall, I find the manuscript to be well written and concise. The authors examine a very important topic. It is my opinion that the manuscript will be an important contribution to the scientific literature on this topic. I do have two major points of criticism, however, that should be addressed before the paper is published.

Thus, I recommend a major revision of the paper with more critical focus on the discrepancies in the pristine Southern Ocean.

We are very grateful to the reviewer for thoroughly reviewing our manuscript and for providing valuable insights to improve it. We also thank the reviewer for highlighting several important studies on Southern Ocean CCN concentrations, which have significantly altered some of our manuscript's main findings and enhanced the overall quality of our paper. We agree with most of the points raised by the reviewer. Below is our point-by-point response to the questions raised:

**Comment 1:** My main point is that the authors neglect several papers that document the seasonal cycle of CCN in the Southern Ocean. The authors correctly cite the fact that in situ data sets are rare, but they seem to have missed several very strong observational studies that could bring light to the seasonal cycle discrepancy they find in the pristine Southern Hemisphere oceans. For instance, data from the Cape Grim observatory have been used to demonstrate the seasonal cycle in CCN in Southern Ocean air masses in papers dating back to the early1990's (Ayers and Gras, 1991) and more recently (Gras and Keywood, 2017) looking at more than 3 decades of data. While the Cape Grim observatory is situated just a few hundred km from mainland Australia, the authors of these papers are careful to use only data that represent pristine Southern Ocean air masses that

have had long trajectories over open water to the southwest. Both papers show a seasonal cycle in CCN that is in striking agreement with the CAM dataset that have a winter minimum in CCN in all the Southern Ocean regions analyzed. While the authors cite the paper by Humphries et al. (2023) to support the CALIOP winter maximum in Southern Ocean CCN arguing that higher winds drive sea salt aerosol, the Humphries et al. paper also shows in situ seasonal cycles from ships over a wide latitude belt extending from Australia to Antarctica that agree boradly with the winter minimum in CCN. This winter minimum extends to low-level clouds as well. McCoy et al. (2015) demonstrate such a seasonal cycle analyzing MODIS cloud data while Mace and Avey (2017) analyze CloudSat data to also show a significant winter minimum in Nd over the Southern Ocean.

It is my opinion that the authors really must address this body of literature since it seems evident that the CAM data set accurately captures the seasonal cycle in the pristine Southern Ocean while the CALIOP data set simply does not. This would imply that the CALIOP retrieval algorithm has serious issues in pristine oceanic regions.

(a) Regarding CCN cycles in Southern Ocean

As pointed out by the reviewer, several in-situ studies have shown a winter minimum in CCN concentrations in the Southern Oceans, which were previously not considered in our manuscript. We agree with the reviewer that the seasonal changes in CCN concentrations in pristine marine environments are better represented in CAMS. In response, we have clarified this in Section 2.2 of the revised manuscript by discussing the literature suggested by the reviewer, along with some additional references. We have also introduced a new domain in our analysis for the Southern Ocean, defined by latitudes 40S-65S and longitudes 30W-180E. Unsurprisingly, the CAMS-derived monthly variations in CCN concentrations for this new domain align with in-situ measurements and are opposite to those from CALIOP.

We then investigated the monthly CCN variations for different aerosol species to determine if these differences persist even in marine-specific aerosol species. Additionally, we incorporated monthly variations in CALIOP-derived aerosol extinction coefficients and CAMS-derived aerosol mass mixing ratios to identify key discrepancies in the original satellite and reanalysis products that may contribute to the observed differences in CCN concentrations over pristine oceans.

To summarize our findings, although the total monthly CCN variations differ between CAMS and CALIOP, the variations in sea-salt CCN from CAMS are surprisingly similar to those of marine CCN in CALIOP, with both exhibiting a winter maximum (see blue lines in Fig. R1.1). However, this winter maximum does not appear in the total CCN cycle in CAMS due to the relatively low contribution of sea-salt CCN compared to other species (primarily sulphates). This pattern is also reflected in the primary satellite and reanalysis products used in CCN calculations, that is the extinction coefficient in CALIOP and mass mixing ratio in CAMS. Interestingly, sea salt aerosols in CAMS and marine aerosols in CALIOP contribute predominantly to the mass mixing ratio and extinction coefficient, respectively, for all oceanic domains (second and fourth column of Fig. R1.2). Indeed, in-situ studies have also shown that sea salt contributes primarily to aerosol mass in pristine oceans (Ayers and Gras, 1991; Gras and Keywood, 2017; Humphries et al., 2023). Given this, sea salt aerosols, due to their coarse size and high scattering properties, may dominate the marine extinction coefficient in CALIOP, leading to similar seasonal variations in their extinction coefficients and mass mixing ratios.

This dominance of sea salt aerosols in CALIOP-derived CCN concentrations poses a problem, as the CCN cycle in CALIOP closely follows sea-salt CCN variations (due to the assumed proportionality between CCN concentrations and extinction coefficient), even though sea salt is not the primary source of CCN in marine environments. In-situ studies have demonstrated that sulphates from biogenic emissions are the primary contributors to CCN in pristine oceans (Ayers and Gras, 1991; Gras and Keywood, 2017; Humphries et al., 2023). Since CALIOP cannot distinguish between sea salt and biogenic emissions, there is a need to separate marine aerosols into these components. However, this separation requires properties like lidar ratios and depolarization ratios for sea salt and biogenic aerosol components, which are currently unavailable. We have highlighted these limitations in CALIOP in the revised manuscript. CAMS, on the other hand, distinguishes between sulphates and sea salt aerosols over oceans, producing a more representative overall CCN cycle in these regions.

That said, we believe that sea salt CCN in CAMS, which exhibits negligible contribution to total CCN in most oceanic domains (see Fig. R1.2), may be underrepresented. This is because sea salt can contribute 8–51% of the total CCN when surface wind speeds are below 16 m/s and up to 100% at higher wind speeds (Fossum et al., 2018).

[revised manuscript text omitted]

(b) Regarding the seasonal $N_d$ cycles

We agree with the reviewer. The winter minimum in CCN is indeed reflected in $N_d$, as demonstrated in previous studies. To address this, we have computed the seasonal $N_d$ cycles for various regional domains using MODIS $N_d$ data (see Fig. R1.3 below). This clarification has been incorporated in Section 2.2 of the updated manuscript. Additionally, Fig. R1.3 has been added to the supplementary section as Fig. S2.

The following text has been added to the revised manuscript between lines 131 and 134:

*"As a result, cloud droplet number concentrations ($N_d$), a parameter sensitive to changes in $n_{CCN}$, also displays a spring–summer maxima and winter minima in pristine Southern Oceans (McCoy et al. (2015); Mace and Avey (2017); see also Fig. S2 in the supplementary material). These seasonal CCN cycles are well represented in CAMS but not in CALIOP."*

[Figure]

*Figure R1.3: Monthly variations in cloud droplet number concentrations ($N_d$) for various regions. Panels (a) to (i) correspond to Northern Hemisphere regions, while panels (j) to (p) represent Southern Hemisphere regions. Datasets from June 2006 to December 2020 are used to generate the monthly climatology.*

**Comment 2:** I am also quite skeptical of the trend analysis presented in this paper. There is very little discussion of the methodology. The instruments being used (CALIOP and MODIS) aged

substantially over the period considered. The authors do not discuss how they have accounted for the aging of instruments and how this has been accounted for in their trend analysis.

As stated in Section A3 of the Methods of our manuscript, we apply Mann-Kendall trends in our analysis. We opted for this method instead of linear trends because (i) it is non-parametric, meaning it doesn't require any assumptions about the data distribution (unlike linear regression, which assumes a normal distribution), and (ii) it is more robust against outliers compared to linear trends.

Fig. R1.4 below shows the Mann-Kendall and linear trends computed for annual time-series of CCN concentrations in CALIOP and CAMS across 16 regional domains. The figure shows that the trend signs in both methods are the same. The differences are in the magnitudes of the trend. Moreover, by just observing the annual time series for the SH oceanic domains in the bottom row (IO, Sat, SEP, and SO), we can point out that CAMS show an increasing trend in CCN concentrations, which are accurately represented by Mann-Kendall trends. Furthermore, similar increasing trends in MODIS aerosol loading over the Southern Hemisphere oceans were also identified by Quaas et al. (2022) using linear regression.

[Figure]

*Figure R1.4: Annual time series of CCN concentrations for 16 regional domains considered in the study. LI and MK in the legend of each panel represent the linear slope and Mann-Kendall slope, respectively.*

We have included Fig. R1.4 in the supplementary material as Fig. S4. The following texts are modified and revised in the updated manuscript between lines 299 and 302 for improved clarity:

*Trends in $n_{CCN}$ and $N_d$ are estimated using the non-parametric Mann-Kendall trend test, as it does not require any assumptions about the distribution of the time series data and is more robust in handling outliers (Mann, 1945; Kendall, 1975). Annual trends computed using linear regression are shown in Fig. S4 in the supplementary material.*

Regarding the ageing of the instruments:

We agree with the reviewer that sensor ageing in spaceborne instruments can affect retrieval quality. However, it is important to note that these sensors undergo regular calibration to detect and correct any significant anomalies in the measurements. For example, the CALIOP team consistently updates their official page (available at https://www-calipso.larc.nasa.gov/products/updates.php, last accessed on 30 October 2024) with information regarding potential issues with the sensor. If any degradation in the sensor's performance is detected, the affected data are not processed or made available to the scientific community. For the time period covered in this study, neither the CALIOP nor MODIS science teams have reported any issues related to sensor ageing. Consequently, we did not account for these effects, and no changes have been made to the manuscript on this matter.

**References**:**

[revised manuscript text omitted]

**Reviewer 2**

This is a review for 'Pristine oceans control the uncertainty in aerosol-cloud interactions' by Choudhury et al. The study looks at two different datasets of vertically-resolved gridded CCN data. The first is a satellite-derived product using lidar. The second is a reanalysis. At the end of the introduction they state " Ultimately, this work aims to establish a benchmark for applying and developing CCN-retrieval algorithms in the context of aerosol-cloud interactions". I will target my review with this in mind. I have two major comments and several specific/minor comments. I want to emphasise though that I really admire the work the authors have put into not only this manuscript, but in generating these datasets in the first place. The suggestions I have are really to expand on the great work already done.

We are very grateful to the reviewer, Dr. Marc Daniel Mallet, for thoroughly reviewing our manuscript and providing valuable insights to improve it. We particularly thank the reviewer for suggesting the creation of a separate domain for the pristine Southern Ocean and for recommending a comparison of the base satellite and reanalysis datasets, which have significantly clarified some of the issues encountered in the previous versions of the manuscript. These suggestions have notably revised some of our manuscript's main findings and enhanced the overall quality of our paper. We agree with most of the points raised by the reviewer. Below is our point-by-point response to the reviewer's comments:

**Major comments:**

**Comment 1:**

**Part-1:**

I understand that the CALIOP and CAMS CCN datasets have their own papers describing how each dataset is produced. I think some of that information needs to be brought over into the discussion in this manuscript, particularly when the different assumptions about the aerosol size distribution, hygroscopicity, and activation diameter could be reasons for differences between the two products. However, the differences in these assumptions also need to be weighed against the potential limitations in the two products as well. For CALIOP, that lies in the fact that CCN concentration doesn't always correlate with aerosol extinction. For CAMS, the sources and sinks of different species could be over- or underestimated. Around Line 95 there is a discussion about the fact that CALIOP-derived CCN concentrations are consistently larger than CAMS for most regions. They attribute this to the fact that the CALIOP retrieval assumes a fixed CCN-activation radius, whereas the CAMS product calculates CCN based on the simulated mass mixing ratios of different species. Can the authors rule out other possible causes?

We are thankful to the reviewer for their suggestion. As the reviewer has pointed out, we have presented a discussion on the potential differences in magnitudes of the CCN concentrations derived using CAMS and CALIOP in the manuscript, highlighting the differences in their retrieval algorithms as potential source for this disparity. We have now expanded the potential reasons that can lead to the differences by highlighting the points raised by the reviewer. The following lines are added in Section 2.1.1 (lines 102-107) of the revised manuscript.

*Additionally, CAMS excludes dust as a potential CCN source, which is accounted for in CALIOP. These differences in the assumptions in CALIOP and CAMS in terms of aerosol hygroscopicity, activation size, and CCN activity may naturally lead to lead to higher concentrations in CALIOP compared to CAMS. Other factors may also contribute to these differences. For example, CALIOP's aerosol extinction coefficient may not correlate well with $n_{CCN}$ in complex aerosol mixtures with varying hygroscopicity (Choudhury and Tesche, 2022). Additionally, inaccuracies in the representation of aerosol sources and sinks in CAMS may bias the derived $n_{CCN}$ (Moore et al., 2013).*

As we address in the second part of the comment, we have revised certain parts of Section 2.2 to highlight the potential causes for low CCN concentrations in CAMS for SH oceanic domains compared to observations. These insights were derived from new comparisons of type-specific CCN concentrations, aerosol extinction coefficient and mass mixing ratios. New Figs. R2.1 and R2.2 are also added. These modifications are discussed in detail in the next response. Here, we have revised the following part of Section 2.2 in lines 150-155:

*Nevertheless, CAMS may significantly underestimate the contribution of sea salt aerosols to oceanic $n_{CCN}$ (third column of Fig. A1), which can be as high as 8–51 % of the total $n_{CCN}$ and may increase to 100 % at higher surface wind speeds (Fossum et al., 2018). This underestimation could stem from an underrepresentation of small-mode sea-salt aerosol mass in CAMS (see fourth column of Fig. A1). Another plausible reason may be the size distribution assumed in CAMS's CCN-retrieval algorithm, which may not accurately represent small-mode sea-salt aerosols. Such factors likely contribute to the observed low $n_{CCN}$ values in CAMS compared to in-situ observations for SH oceanic domains (see Fig. 2b).*

**Part 2:**

I'm not suggesting that the authors go and do a whole new study or change anything drastically. But I do wonder if a comparison of the aerosol extinction from CALIOP and calculated aerosol extinction from CAMS might be a fairer comparison, or at least useful in diagnosing the causes of the differences between the two products. At the very least, some discussion should be added on this point, both when discussing the spatial and seasonal differences between the two products.

We agree with the reviewer. In the revised manuscript, we have incorporated two new comparisons that have significantly enhanced our understanding of the differences in the datasets: (i) a comparison between marine and non-marine CCN concentrations in CALIOP with sea-salt and non-sea-salt CCN in CAMS, and (ii) a similar comparison where CALIOP aerosol extinction coefficients are compared with CAMS mass mixing ratios instead of CCN. We use mass mixing ratios from CAMS as they serve as the basis for the derived CCN concentrations and are readily available for download from the official CAMS website.

Before discussing the findings of the comparison, we briefly note additional studies and results crucial for understanding the comparisons. In Fig. 3 of the manuscript, we showed that the CCN seasonal cycles in CALIOP are opposite to those in CAMS, with CALIOP exhibiting a winter maximum and summer minimum. As highlighted by the first reviewer, several studies have rigorously examined CCN cycles in pristine oceans. The key conclusions from published in-situ studies are:

(i) CCN concentrations in pristine oceans exhibit a winter minimum and a spring-summer maximum (Gras, 1990; Ayers and Gras, 1991; Gras and Keywood, 2017; Humphries et al., 2023). This pattern is also observed in cloud droplet number concentrations (Nd; McCoy et al., 2015; Mace and Avey, 2017). CAMS accurately captures this seasonal cycle, whereas CALIOP does not.

(ii) CCN in pristine oceans primarily originates from sulphates produced by biogenic emissions (Gras, 1990; Ayers and Gras, 1991; Gras and Keywood, 2017; Humphries et al., 2023), which follow seasonal patterns in the total CCN concentrations (Lana et al., 2011; Revell et al., 2019). In contrast, sea salt aerosols predominantly contribute to aerosol mass in marine environments (Humphries et al., 2023).

We now consider these published literature and the new results from the comparison study suggested by the reviewer. We find that although the total monthly CCN variations are opposite between CAMS and CALIOP, the variations in sea-salt CCN from CAMS are surprisingly similar to those of marine CCN in CALIOP, with both exhibiting a winter maximum (see blue lines in Fig. R2.1 below). However, this winter maximum in CAMS's sea salt CCN does not appear in the total CCN cycle due to the substantially low concentrations of sea-salt CCN compared to other species (primarily sulphates; see Fig. R2.2). This pattern is also reflected in extinction coefficient in CALIOP and mass mixing ratio in CAMS. Sea salt species in CAMS and marine species in CALIOP contribute predominantly to the mass mixing ratio and extinction coefficient, respectively, for all oceanic domains (second and fourth column of Fig. R2.2). This is in accordance with the in-situ studies, which show that sea salt contributes primarily to aerosol mass in pristine oceans. Given this, sea salt aerosols, due to their coarse size and high scattering, may dominate the marine extinction coefficient in CALIOP, thereby leading to similar seasonal variations in extinction coefficients in CALIOP and mass mixing ratios in CAMS.

This dominance of sea salt aerosols in CALIOP-derived CCN concentrations for marine aerosols poses a problem, as the CCN cycle in CALIOP closely follows sea-salt CCN variations, which is not the primary source of CCN in marine environments. This may also contribute to the high CCN concentrations observed in CALIOP in marine environments compared to CAMS and in-situ observations. As shown in in-situ studies, sulphates from biogenic emissions are the primary contributors to CCN in pristine oceans (Ayers and Gras, 1991; Gras and Keywood, 2017; Humphries et al., 2023). Since CALIOP cannot distinguish between sea salt and biogenic emissions, there is a need to separate marine aerosols into these components. However, this separation requires properties like lidar ratios and depolarization ratios for sea salt and biogenic aerosol components, which are currently unavailable. We have highlighted these limitations in CALIOP in the revised manuscript. CAMS, on the other hand, distinguishes between sulphates and sea salt aerosols over oceans, producing a more representative CCN cycle in these regions. That said, we believe that sea salt CCN in CAMS, which shows negligible contribution to total CCN in most oceanic domains (see Fig. R2.2), may be underrepresented. Sea salt can contribute 8–51% of the total CCN when surface wind speeds are below 16 m/s and up to 100% at higher wind speeds (Fossum et al., 2018).

[revised manuscript text omitted]

**Comment 2:** It really stands out to me that high latitudes have been excluded from this study, even though there is global data from both the CALIOP and CAMS products. Aside from a small part of the Southern America domain, the regional domains used for this study extend only as far south as 40S. There are in situ observations from the Arctic, the Antarctic continent, and across the Southern Ocean that could be used for comparison as well. The Southern Ocean south of 40S is arguably the most pristine region on the planet, and also very important in terms of aerosol-cloud-radiation interactions. I would not be alone in thinking that it is disappointing that this region has been excluded from the analysis, especially given the title of the study.

We thank the reviewer for pointing this out. Considering the significance of the pristine Southern Ocean for aerosol-cloud interactions, we have introduced a new domain in our analysis: the Southern Ocean, defined by latitudes 40°S-65°S and longitudes 30°W-180°E. However, we restrict our analysis in this work to latitudes within 65 degrees and therefore do not consider polar regions. MODIS observations at such high latitudes are subject to biases due to high solar zenith angles (Grosvenor and Wood, 2014; Grosvenor et al., 2018), making accurate retrievals of cloud and aerosol properties challenging. Satellite-based studies estimating radiative forcing due to aerosol-cloud interactions often exclude these regions due to such retrieval issues (Gryspeerdt et al., 2019; Hasekamp et al., 2019; Forster et al., 2021; Gryspeerdt et al., 2023). Due to this reason, such observations are not assimilated into CAMS (Inness et al., 2019), making it rely mostly on the underlying model to estimate aerosol mass mixing ratios. Moreover, CALIOP retrievals, particularly its aerosol classification algorithm, have not been thoroughly validated at high latitudes. For instance, the aerosol-species-specific climatology of CCN concentrations shown below in Fig. R2.3 for Antarctica shows the presence of polluted continental aerosols along the coastline (see Fig. R2.3b), which is opposite to what one would expect for such pristine environments.

These factors complicate the validation of derived CCN concentrations and their comparisons with $N_d$, as the base products themselves may be associated with uncertainties that have not yet been quantified. Therefore, we exclude polar regions from our comparison study. We have now modified the following sentence in Section A3 of our manuscript in lines 289-291:

*CCN, cloud, and precipitation parameters are considered between latitudes of $65^oN$ and $65^oS$. Data at higher latitudes are not considered due to the uncertainties associated with MODIS observations at high solar zenith angles (Grosvenor and Wood, 2014; Grosvenor et al., 2018) and the lack of validation for CALIOP retrievals at these latitudes.*

[Figure]

*Figure R2.3: Aerosol-type-specific climatology of CCN concentrations (in cm⁻³) over Antarctica (latitude south of 65°S) derived from CALIOP for (a) Marine, (b) Polluted Continental, (c) Smoke, (d) Dust and (e) Clean Continental aerosols.*

**Specific comments:**

1. Figure 3. The differences in the shape of some of these seasonal cycles is stark (i.e. Indian Ocean, Southeast Pacific, South Atlantic). The authors mention that ocean CCN are influenced by more than sea spray aerosol but do not go much further than that. There is certainly enough in situ data to confirm there is an increase in austral summertime biogenic CCN around the Southern Ocean. Furthermore, the South Atlantic is strongly influenced by biomass burning during the dry season. CAMS seems to at least represent these expected seasonal cycles in these regions, where-as the CALIOP product does not. I'd encourage the authors to discuss these limitations in the context of the in situ observational studies that have taken place in these regions. I do think this is important as these products could end up being used to either evaluate or serve as climatologies in future modelling efforts.

We agree with the reviewer. As highlighted in the previous responses, we have significantly revised our manuscript by discussing published in-situ studies that agree with the seasonal cycles in CAMS and not with CALIOP. Furthermore, we have now identified the issue in CALIOP, in which the CCN concentrations, being proportional to the extinction coefficient (mostly comes from coarse sea salt particles), follow the seasonal variations in sea salt aerosols. We have further highlighted the need for separating the extinction coefficient of marine aerosol species into contributions from sea salt and biogenic aerosols, for improving the representativeness of CALIOP-derived CCN concentrations in pristine marine environments.

2. I think the recommendations for future work and conclusions could be strengthened a bit. Although there are perhaps more existing (and planned!) in situ CCN measurements in the remote regions of the Southern Hemisphere oceans than implied in the manuscript, I agree that the sources (also sinks) of CCN do need to be better represented in CAMS (models in general). But can the

authors comment on the suitability of satellite-based products of CCN in these regions? CCN can vary considerably due to variability in aitken and accumulation-mode biogenic aerosol, that could be nearly independent of the sea spray aerosol that determines the larger accumulation and coarse mode that aerosol extinction is sensitive to. How can that limitation be overcome? Maybe it cannot, and a hybrid approach with machine learning or earth system modelling is needed. I think adding this type of discussion is useful for determining when and where each product could be considered a "truth".

Thank you for your suggestions. Based on the additional results incorporated in the revised manuscript, we have modified our conclusion section accordingly. First, we highlighted the limitations in CALIOP (in representing seasonal CCN cycles) and CAMS (in representing the annual trends), emphasizing the need for caution when using these datasets in the pristine oceans of the Southern Hemisphere. We also identified a key area of improvement for CALIOP, stressing the need to separate its marine aerosol components into sea salt and biogenic aerosols. Additionally, we discussed the necessity of increased observations in these regions, which are essential for enhancing the representativeness of CAMS aerosol inventories. Finally, we included machine learning as a promising tool for estimating global CCN datasets. These updates have been reflected in the conclusion section (lines 186-207) of the manuscript as:

*The closure study presented here shows good consistency between the independent CALIOP and CAMS global $n_{CCN}$ datasets in continental environments. However, significant discrepancies emerge over most pristine oceans, not only in $n_{CCN}$ climatology but also in their monthly and annual variations. While the seasonal cycles of oceanic $n_{CCN}$ derived from CAMS largely align with previous in-situ observations (Gras, 1990; Ayers and Gras, 1991; Gras and Keywood, 2017; Humphries et al., 2023) and the variations in $N_d$, CAMS likely underestimates the contributions from sea salt $n_{CCN}$. In contrast, the seasonal cycles in CALIOP are not representative, likely due to its inability to resolve marine $n_{CCN}$ into sea salt and sulphate (from biogenic emission) components.*

*The results, however, are completely opposite for annual trends in $n_{CCN}$ and $N_d$. While trends in CAMS and CALIOP generally agree across most NH regions, they diverge significantly in the SH. CALIOP consistently shows a declining $n_{CCN}$ trend in these regions, which aligns with previous reports (IMO, 2019; Gryspeerdt et al., 2019; Quaas et al., 2022) and the decreasing trend in $N_d$, while CAMS exhibits an anomalous increasing $n_{CCN}$ trend over SH oceans. This geographically limited disagreement, restricted to pristine oceans with limited in-situ measurements, raises questions about the adequacy of aerosol inventories used by CAMS in SH oceans, a known issue in climate models (Moore et al., 2013). These discrepancies in cloud-rich pristine oceans are particularly concerning because cloud properties in these regions are highly sensitive to even small perturbations in aerosol concentrations (Moore et al., 2013; Gryspeerdt et al., 2023).*

*Caution should therefore be taken when using these $n_{CCN}$ datasets in the pristine oceans of SH. Future research efforts should focus on first separating the sea salt and biogenic components of marine aerosols in CALIOP, and second, on accurately quantifying the sources and sinks of CCN and their long-term cycles in remote SH oceans for further improving the representativeness of aerosol inventories in CAMS. An alternative approach could involve the development of advanced data-driven techniques to derive global CCN dataset (Redemann and Gao, 2024). These efforts are crucial to refine the global $n_{CCN}$ datasets and ultimately to reduce the uncertainties in $ERF_{ACI}$.*

**Minor comments:**

1. How do these two CCN-products compare to MODIS Nd retrievals? It would be really insightful to show the correlation coefficient between Nd and CCN for both datasets (similar plot to Figure A2a).

Thank you for your suggestion. We have compiled a plot similar to the joint histogram of Gryspeerdt et al (2023) (see Fig. R2.4 below) to show the correlation between $N_d$ and $n_{CCN}$. Interestingly, we find that the $N_d$-$n_{CCN}$ susceptibility ($\frac{d \ln N_d}{d \ln n_{CCN}}$), estimated as the linear slope to the mean variations in $N_d$ with $n_{CCN}$ (red line in Fig. R2.4), falls within the previous observational estimates by Gryspeerdt et al (2023) (between 0.11 and 0.36) for both CALIOP and CAMS. However, since the datasets are on a monthly scale, we expect their correlations to be influenced by their mean variability, which may obscure their causal relationship. As a result, we do not explicitly discuss these correlations in the manuscript. Though we have included Fig. R2.4 in the supplementary materials as Fig. S9 for readers who may be interested in exploring these correlations.

Instead, we use $N_d$ as a proxy to approximate the monthly variations in $n_{CCN}$ in Section 2.2 of the manuscript. Given the intrinsic positive correlation between $N_d$ and $n_{CCN}$, we expect the seasonal cycles in $n_{CCN}$ to be similar to $N_d$, particularly over oceans, which are mostly cloudy. Indeed, we find the seasonal $N_d$ cycles over pristine oceans (see Fig. R2.5) to be akin to the in-situ reported variations in $n_{CCN}$, which are well represented in CAMS and not in CALIOP. Fig. R2.5 is now included as Fig. S2 in the supplementary.

[Figure]

*Figure R2.4: Global aggregated joint histograms depicting the conditional probability of occurrence of $N_d$ for a given $n_{CCN}$. m represents the $N_d$-$n_{CCN}$ susceptibility term estimated as the slope of the linear regression to the mean $N_d$ (red line) for every $n_{CCN}$ interval. Monthly $N_d$ and $n_{CCN}$ datasets between June 2006 and December 2020 are used.*

[Figure]

*Figure R2.5: Monthly variations in cloud droplet number concentrations ($N_d$) for various regions. Panels (a) to (i) correspond to Northern Hemisphere regions, while panels (j) to (p) represent Southern Hemisphere regions. Note the separate y-axes for CALIOP (left) and CAMS (right). Datasets from June 2006 to December 2020 are used to generate the monthly climatology.*

2. I would like to see a version of Figure 1 broken down into seasons in the appendix or supplementary material.

The figure is now added to the supplementary material as Fig. S10.

3. I would like to see Figure A1 incorporated into Figure 2. The regional domains are used and referenced a lot, so I'm not sure if the map showing what those domains are should be in the Appendix.

Figure A1 is now merged with Figure 2 of the updated manuscript.

4. The time period the trends are calculated over in Figure 4 and 5 and A3/A4 should be in the caption or legend.

Time ranges are included in all plots in the manuscript.

5. Figure A2. There is something that has gone wrong with the plotting for panel b. The coast lines do not match up with the cloud cover pixels properly. You can see this clearly at the bottom of South America, Africa, Australia and New Zealand. Can the authors fix this but also really check the plotting issue isn't also there in Figures 1, 4, A3, A4, A5.

Thank you for pointing out this error. There was a bug in the code used to generate the plot. The issue was with the latitude array, which did not correspond to the CERES grid. We have now

rectified this in the updated manuscript. Other figures that display global maps were not affected by this issue.

6. Lat and lon grid lines would be very much appreciated on the map plots.

Lat and lon grids and their labels are now included in all global map plots in the manuscript.

7. Line 71: "below 45S" should be "south of 45S" as "lower" latitude is closer to the equator than a "higher" latitude

Corrected.

8. Line 81. I'm not quite sure what is meant by land-ocean gradients of e.g 65%.

We apologize for the confusion. What we meant is that the land concentrations are 65% higher than those over the ocean. We have now revised the sentence (between lines 82 to 84) as follows:

[revised manuscript text omitted]

---

## Author Response (AR3)

Dear Dr. Timothy Garrett,

Thank you for your suggestions. We have made the required changes to the manuscript. The following updates are reflected in the revised version (newly added text highlighted in red and deletions in blue):

Line 13: Now reads *"A closure study of trends in CCN and cloud droplet concentrations suggests that dust-influenced and pristine-maritime environments are primary regions that limit our current understanding of CCN-cloud-droplet relationships."*

Line 58: $N_d$ is now defined earlier in the introduction as *"... reconciling not only their variability across diverse spatio-temporal scales but also their co-variability with relevant cloud properties, such as cloud droplet number concentration ($N_d$)."*

Additional minor corrections:

Line 56: The supersaturation value used for comparing CCN concentrations is now introduced earlier in the manuscript: *"Here, we conduct a closure study between the two independent novel $n_{CCN}$ datasets for a maximum supersaturation of 0.2 %, reconciling ..."*

Line 64: Modified from *"We first compare the spatial variations in $n_{CCN}$ climatology at a supersaturation of 0.20 % for altitudes ..."* to *"We first compare the spatial variations in $n_{CCN}$ climatology for altitudes ..."*

Acknowledgement section: Funding sources are updated.

Thank you for your consideration.

Best regards,

Goutam Choudhury